# Effects of Nitrogen Forms on the Growth and Nitrogen Accumulation in *Buchloe dactyloides* Seedlings

**DOI:** 10.3390/plants11162086

**Published:** 2022-08-10

**Authors:** Lizhu Guo, Huizhen Meng, Ke Teng, Xifeng Fan, Hui Zhang, Wenjun Teng, Yuesen Yue, Juying Wu

**Affiliations:** 1Institute of Grassland, Flowers and Ecology, Beijing Academy of Agriculture and Forestry Sciences, Beijing 100097, China; 2College of Life Sciences, Northwest University, Xi’an 710069, China

**Keywords:** nitrogen forms, biomass allocation, nitrogen allocation, ^15^N, *Buchloe dactyloides*

## Abstract

Buffalograss [*Buchloe dactyloides* (Nutt.) Engelm.] has become the most widely cultivated warm-season turfgrass in northern China because of its low-maintenance requirements. Nitrogen (N) can be applied to plants in a range of formulations. However, preference of nitrogen uptake and the effects of N form on plant growth and nitrogen accumulation has not been established in buffalograss. In this study, we evaluated the effects of different inorganic nitrogen forms (NO_3_^−^-N, NH_4_^+^-N, and NO_3_^−^-N: NH_4_^+^-N = 1:1) on growth and nitrogen accumulation in buffalograss seedlings. Results showed that supply of three N forms significantly increased buffalograss seedlings growth, biomass, and N contents of all plant organs compared with the seedlings receiving free nitrogen. Plants achieved better growth performance when they received nitrate as the sole N source, which stimulated stolon growth and increased the biomass of ramets, spacers, and aboveground and total plant biomass, and also allocated more biomass to ramets and more N to spacers. Meanwhile, those plants supplied with the treatment +NH_4_NO_3_ displayed a significantly greater N content in the ramet, ^15^N abundance, and ^15^N accumulation amount in all organs. These data suggest NO_3_^−^-N supplied either singly or in mixture increased vegetative propagation and thus facilitates buffalograss establishment. However, applications of ammonium caused detrimental effects on buffalograss seedlings growth, but +NO_3_^−^ could alleviate NH_4_^+^-induced morphological disorders. Thus, recommendations to increase vegetative propagation and biomass accumulation in buffalograss seedlings should consider increasing NO_3_^−^-N in a fertility program and avoiding applications of nitrogen as NH_4_^+^-N.

## 1. Introduction

Nitrogen (N) is an essential nutrient for plant growth and development in terrestrial ecosystems [1] and plant ecologists increasingly recognize that plants can capture nitrogen in a variety of different chemical forms, ranging from inorganic forms such as nitrate (NO_3_^−^) and ammonium (NH_4_^+^) to different amino acids and more complex organic N [2,3]. Due to differences in environmental conditions, plant species, and the nutritional characteristics of N sources, plants have adapted to different N forms during long-term evolution and may show optimized growth with specific N forms [4]. For example, *Phyllostachys edulis* [5], blueberry [6], and rice [7] show improved growth with available NH_4_^+^, whereas some crops and early successional pioneer species prefer NO_3_^−^ [4].

Nitrate and ammonium as different forms of nitrogen nutrients impact differently on some physiological and biochemical processes in higher plants [8,9,10,11]. French bean (*Phaseolus vulgaris*) with solely nitrate (NO_3_^−^) supply had higher root (36%) and shoot dry matter (11%) than plants with solely ammonium (NH_4_^+^) supply [12]. Relative to urea or ammonium, exclusive supply of nitrate increased tiller number in hydroponically grown barley plants [13]. Applications of nitrogen as ammonium caused strong detrimental effects on pecan (*Carya illinoinensis* Wangenh. K. Koch) growth, characterized by decreased total biomass and root growth [14]. Compared with nitrate, biomass of wheat was reduced by 54% or 85% under low or high ammonium [15]. NH_4_^+^-N stimulated tuber swelling of potato plants (*Solanum tuberosum* L.), while NO_3_^−^-N stimulated the branching of stolons and stems, as well as shoot growth [16]. Apart from the influence of N form on the ecophysiological characteristics and dry matter partitioning between shoot and root, it was also found that the nitrogen uptake and utilization also differ with N forms [17,18]. Nitrogen use efficiency (NUE) is an established metric used to benchmark N management [3,10,19]. NUE decreases with increased N supply, but it can also be dependent on the form of N taken up and utilized [20,21]. Thus, to maximize NUE, a thorough understanding of the effects of the different forms of N on plant growth and development is required [11]. Furthermore, like plant functional traits (e.g., leaf N, or leaf mass per area), the allocation of nutrients among organs can be considered as another plant trait that may reflect interactions between plants and their environments per se [22]. To maximize plant growth and maintain the optimal metabolic activities, plants need to balance the allocation of nutrients across organs under different biotic and abiotic stresses [23]. The nutrient concentrations of different organs are related to organ function, organ growth and turnover rates, and plant growth forms [24,25]. However, current knowledge of plant nutrient allocation strategies mainly comes from crops and economic forest tree species [5,6,14,15], with little knowledge from turfgrasses.

Buffalograss (*Buchloe dactyloides* (Nutt.) Engelm.) is a warm-season grass for low-maintenance lawns. This species is highly tolerant to drought, heat, and other abiotic stresses, and can adapt to a variety of soil conditions [26,27]. Drought stress and water shortage is now a critical issue worldwide, and drought-tolerant turfgrass species such as buffalograss are being explored for use in the landscape to help conserve water supplies [28]. Furthermore, buffalograss is also commonly used for roadside revegetation [29,30], canal slope protection [31], phytostabilization of mine tailings [32,33], and airport land-cover [34]. Playing multiple roles, buffalograss has become the most widely cultivated turfgrass in northern China. However, plant establishment can be difficult, particularly in harsh soil conditions, due to low organic carbon content, nutrients, and moisture. Thus, amendments are required, such as fertilizer to facilitate vegetative propagation that could facilitate rapid land-cover. Moreover, fertility applications have the potential to reduce the time to establishment without incurring significant cost [26]. Previous studies showed that nitrogen stimulates buffalograss growth and biomass production [26,35,36]. However, studies evaluating the effects of nitrogen form on growth, development, and nutrient uptake in buffalograss are currently lacking. Such studies in turf plants are problematic in that, under field conditions, it is difficult to control nutrient levels and variations in soil properties.

In this study, sand culture was used, which allowed strict control of nutrient supply and precise measurement of nutrient uptake. The aims of this study were to determine if buffalograss exhibits an uptake preference for different N forms, and to evaluate the effects of the N forms on growth characters, biomass accumulation, and uptake of N and N allocation on the whole plant level. Such information is useful in optimizing fertilization schemes and our understanding of buffalograss nutrition. Deciphering the response of buffalograss seedlings to different nitrogen form will help maximize buffalograss establishment in various environments. It will also provide solutions for addressing the major issues of pollution and costs related to N fertilizer use that threaten agricultural and ecological sustainability.

## 2. Results

### 2.1. Effect of Nitrogen Form on Growth Characters

N supply significantly affected most growth parameters except plant height (Figure 1A). Compared with CK, the number of tillers (Figure 1B), number of stolons (Figure 1C), stolon length (Figure 1D), and pitch number of stolon (Figure 1E) in buffalograss seedlings were significantly increased by the nitrogen treatment, and the effects of the separate treatment +NO_3_^−^ (N1) was significantly greater than those of the separate treatment +NH_4_^+^ (N2) and the treatment +NH_4_NO_3_ (N3). Conversely, root average diameter under the separate treatment +NO_3_^−^ was significantly lower than the other two N forms treatments (Figure 1H). For spacer length (Figure 1F), there were no significant differences among the three N forms, but all N forms treatment increased the length of spacer. In addition, the separate treatment +NO_3_^−^ significantly increased the root surface area (Figure 1G) and root length (Figure 1I), but the other two treatments had insignificantly greater effects than CK.

### 2.2. Effect of N Forms on Biomass Allocation

Compared with CK, the separate treatment +NH_4_^+^ and +NO_3_^−^ increased the biomass in the root (belowground part) and ortet, but the +NH_4_NO_3_ treatment was insignificantly greater than CK (Figure 2A,D). Additionally, spacer biomass (Figure 2B), ramet biomass (Figure 2C), aboveground part biomass, and whole-plant biomass (Figure 2E) were significantly increased by the three N form treatments, and they were significantly higher with the +NO_3_^−^ treatment than with the +NH_4_^+^ treatment. Meanwhile, N form had significant effects on biomass allocation of buffalograss (Figure 2F). Overall, the biomass allocations of roots and ortets under CK were significantly greater than those under N-supply treatments. However, the ramet biomass and spacer biomass displayed an opposite trend. Both were higher significantly under N forms treatment under CK. There was no significant difference of biomass partitioning in spacers among the three N form treatments, but a significant difference in ramets was observed when between the +NO_3_^−^ and +NH_4_^+^ groups.

### 2.3. Effect of N Form on Nitrogen Contents and Allocations

N supply, regardless of its form, significantly increased N concentration of all organs (Figure 3A), and there was no significant difference among the three different N forms except for the N content in ramets. The N contents differed significantly between the organs (*p* < 0.001). The N content was higher in the ramets and ortets and lower in the roots and spacers. N partitioning in the four organs is shown in Figure 3B. The CK group without N application retained a larger proportion of N in the root and ortet than in the N addition groups. For the allocation of N to spacer, different N form treatments showed significantly higher values than the CK treatment, and the highest was found in the +NO_3_^−^ treatment group. The +NH_4_NO_3_ treatment resulted in significant higher N allocation in the ramets than the CK and +NH_4_^+^ treatments, but was similar to the +NO_3_^−^ treatment. Organ, treatment, and their interactions had a significant influence on N contents (Table 1). Treatments had no significant effects on N allocation, but significant organ * treatment interactions were observed on N allocation.

### 2.4. Effect of N Form on ^15^N Enrichment and ^15^N Distribution

All compartments of the labelled plants were enriched in ^15^N (atom%), with the lowest found in the root and highest in the ortet (Table 2). Besides, variations in ^15^N abundance showed significant differences in ramets among the three N form treatments. Specifically, the ^15^N value of ramets in the +NH_4_NO_3_ treatment was the highest (give P). The percentages of N derived from fertilizer-N (Ndff, %) in three organs (root, spacer, and ortet) were insignificantly different among three treatments with N-supply forms, while a significant difference of Ndff was found in the ramet. The values of Ndff, ^15^N accumulation amount, and ^15^N distribution in the ortet were significantly higher than in the other three organs in all treatments. In addition, there was no significant difference of ^15^N accumulation amount and ^15^N distribution among all the treatments. 

## 3. Discussion

Previous research has focused on the influence of nitrogen source or nitrogen rate applied on buffalograss establishment [26,35,37]. Optimal nitrogen forms for establishment of buffalograss from seed was lacking. Our study shows that buffalograss seedlings prefer N in its nitrate form, because exclusive supply of nitrate increased their vegetative propagation, thus facilitating buffalograss establishment. This can also be corroborated by the fact that the content of nitrate N was significantly lower in the field soil cultivating buffalograss (data not shown). We also found root growth inhibition by NH_4_-N in buffalograss, which was like Creeping Bentgrass [38,39]. Generally, roots constitute the first NH_4_^+^ sensor, and the initial signals of NH_4_^+^ toxicity appear at the root level with a severe modification of the root system architecture, including shorter primary root systems, the inhibition of root elongation, etc. [40,41]. Although NH_4_^+^ can be used as a sole N source and an essential intermediate, it can also result in toxicity symptoms in many plant species, especially when high NH_4_^+^ concentrations are provided as a sole N source [42]. A previous study showed that shoot and root growth is significantly suppressed in cucumber grown with 10 mmol/L of NH_4_^+^ [43]. Similar results have also been found in Arabidopsis, barley, tomato, and beans after high NH_4_^+^ treatments [40,44,45,46], but the NH_4_^+^concentration in our study was moderate. This suggests that buffalograss seedlings could be highly sensitive to the exclusive supply of ammonium nitrogen.

Typical N deficiency symptoms (diminutive and chlorotic canopies) were particularly evident in the CK group, indicating that buffalograss on the field with N deficiency would not grow rapidly, which is expected to inhibit the establishment of buffalograss. What is more, we found that the form of N acquired affected aboveground biomass more than the below-ground part of buffalograss seedlings. This may be the result of increased nitrogen assimilation and allocation to the aboveground part. In aboveground organs, leaves accumulate most of the N in the plant, and about half the total leaf N is used for photosynthetic activities. Thus, the photosynthetic apparatus is the largest sink of N in the plant [47,48]. Moreover, the form of N available exerts a strong influence on plant biomass partitioning, and this response has been interpreted as plants maximizing resource capture by allocating resources to the tissue in which the limiting resource is acquired [2]. Arguably, such optima must depend on the actual chemical form(s) of N present for uptake, because plant N acquisition is mainly limited by processes through which N sources come into contact with root surfaces: mass flow induced by transpiration, diffusion, and interception [49]. Plants adjust to variations in resource availabilities by variable partitioning to root and shoot growth, also known as ‘above- and below-ground biomass allocation’ [50,51,52]. Plants growing on infertile, low N soils are reported to have a higher root mass fraction than plants growing on more fertile and N rich soils [53,54]. Our results showed that buffalograss seedlings under the N-free treatment (CK) partitioned significantly more biomass belowground than the N supply treatments, which is in good agreement with other literature. Deficient nitrogen supply caused increments of biomass partitioning to roots in order to increase root growth, and further increase the absorption of nitrogen to meet the nutritional needs of plants. We also found that the proportion of the ortet mass in CK group was the greatest, but this was mainly because buffalograss seedlings hardly reproduce vegetatively under N-free treatment. Moreover, there were no significant differences of biomass allocation to spacer among three N forms, but the separate treatment of +NH_4_^+^ resulted in significantly lower biomass partitioning in ramets than the separate treatment +NO_3_^−^. Our results suggest NH_4_-N appears to reduce aboveground biomass allocation more than NO_3_-N; this was observed since buffalograss seedlings grown with a supply of exclusively ammonium produced a lower number of stolons, shorter root length, and lower ramet biomass.

Since different plant organs perform different functions in plant growth, the requirements of N will differ [23,24,25]. We found that the N contents of ramets and ortets were significantly greater than those of roots and spacers, which could be the result of leaves containing a high proportion of chloroplasts generally containing more N per unit dry mass than either stems or roots [55]. We also found that N concentration of all organs with N supply were greater than without N, and there was no significant difference among the three different N form applications except for the N content in ramets, which was partly related to the partitioning, as the N in ramets was the greatest among all organs. Furthermore, preferential N allocation to spacer formation is favorable to vegetative propagation of buffalograss seedlings, thus contributing to its establishment in new habitats under the separate treatment +NO_3_^−^.

The use of ^15^N as a tracer is a common means of investigating N uptake dynamics [56,57]. The root represents the compartment with the lowest atom% ^15^N and Ndff% in buffalograss seedlings; this can be explained by the role of roots. During the early-period growth of plants, the N in the culture substrate is absorbed by roots and transported to the developing shoots by the xylem; thus, roots are characterized by low nitrogen concentration. The atom% ^15^N, Ndff, ^15^N accumulation, and the distribution of ^15^N in ortets across all the treatments showed the highest value, followed by ramets, spacers, and roots, which is consistent with previous work showing N is partitioned rapidly into aboveground photosynthetic tissues [58]. Although ^15^N (atom%) tended to be more concentrated in the ortet, the ramet appeared to use the ^15^N more efficiently in biomass production, leading to higher ^15^N inventories despite generally lower ^15^N tissue concentrations. Moreover, we also found that the ^15^N accumulation in vegetative propagules (spacer and ramet) under the separate treatment +NH_4_^+^ was lower than that under the separate treatments +NO_3_^−^ and +NH_4_NO_3_, which would be the reason why applications of nitrogen as ammonium cause detrimental effects on seedling growth, characterized by decreased total biomass and root growth. A growing number of studies have shown that plants supplied with mixed nitrogen sources have more benefits than those supplied with a single NH_4_^+^ or NO_3_^−^ source [59,60]. However, our study showed that buffalograss seedling performance on mixed nitrogen was less than the performance on equal amounts of nitrate-type nitrogen, and partial application of NO_3_^−^ to NH_4_^+^ nutrition can alleviate NH_4_^+^-induced morphological disorders.

## 4. Materials and Methods

### 4.1. Plant Material and Experimental Design

Buffalograss ‘Texoka’ seedlings were grown from pre-germinated seeds in plastic germination boxes and kept in a phytotron under the following day/night conditions: temperature, 25/18 ± 1 °C; relative humidity, 60/80 ± 10 %; photoperiod, 16/8 h; and photosynthetic photon flux density (PPFD), 450 µmol m^−2^ s^−1^ at the top of plastic boxes. The seedlings were watered one time daily during the photoperiod. Seedlings were grown without the addition of nutrients (deionized water only) for one week after germination, then transplanted to soil substrate for 8 weeks, at which point they had an average dry weight of 98.3 mg per seedling. A total of 48 healthy seedlings with similar sizes were chosen and the root systems were washed under tap water to remove soil and surface-dried with filter paper. The seedlings were then replanted in 6 L plastic pots (one seedling per pot) filled with sand in the greenhouse. The sand contained no nutrient addition. Afterwards, the seedlings were watered with deionized water for about one week. The sand culture experiment started on 10 November 2020 and plants were harvested on 2 March 2021.

The experiment contained four treatments: (1) no nitrate and no ammonium (-NH_4_NO_3_, CK); (2) nitrate and no ammonium (+NO_3_^−^, N1); (3) ammonium and no nitrate (+NH_4_^+^, N2); and (4) ammonium and nitrate (+NH_4_NO_3_, N3). Each treatment had 12 plants. In the four different treatments, seedlings were watered with nutrient solution containing differing nitrogen forms. In the nutrient solution nitrification inhibitor, dicyandiamide (DCD, C_2_H_4_N_4_) was used to avoid NH_4_^+^ converting rapidly to nitrate (NO_3_^−^) [61], while the nitrification inhibitors could perform the denitrification efficiently [62,63]. Thus, the ammonium and nitrate could co-exist with little or no conversion. Plants in each treatment groups were watered with 500 mL nutrient solutions every 7 days (Table 3). The main nutrient contents in the nutrient solution, such as sulfur, phosphorus, potassium, calcium, and magnesium were the same among all treatments. The pH of the nutrient solutions was adjusted to 6.0 with HCl or NaOH. Pots were randomized every week to eliminate location effects.

### 4.2. Growth Measurements

Plant height, spacer length, and stolon length were measured with a ruler (cm). The number of tillers, stolon branches, and pinch number of the stolon were counted. Root morphology, including root total length, average diameter, and root surface area, was determined using an automatic scanning apparatus (EPSON color image scanner LA1600+, Toronto, ON, Canada) equipped with WinRHIZO 2007 software (Regent Instruments, Quebec, NA, Canada).

At the end of the experiment, five seedlings were harvested randomly in each treatment. The plants were divided into ortets, spacers, ramets, and roots (Figure 4). Biomass samples were dried (65 °C, 48 h) to a constant weight and weighed. The dry masses of ortets, spacers, ramets, and roots were measured. The total dry mass, aboveground dry mass, belowground dry mass, and biomass fractions of the whole plant biomass allocated to ortets, ramets, spacers, and roots were calculated.

### 4.3. Nitrogen Content Determination

After drying, samples were ground to powder. Total N concentrations in ortets, spacers, ramets, and roots were determined sequentially by a FLASH 2000 elemental analyzer (ThermoFisher Scientific, Waltham, MA, USA). N partitioning in the whole plant between ortets, ramets, spacers, and roots was calculated.

### 4.4. Nitrogen Isotope Composition Determination

In the ^15^N tracer treatment, performed on 25 January 2021, five seedlings were randomly chosen from each N-supply treatment. A total of 30 mg ^15^NO_3_^−^-N or ^15^NH_4_-N per plant was added as Ca(^15^NO_3_)_2_·4H_2_O and K^15^NO_3_ or ^15^NH_4_CL to each treatment (99 atom% ^15^N, Shanghai Research Institute of Chemical Industry, Shanghai, China); the amount applied within the N3 treatment was 60 mg of ^15^N. Then, one month after the application of ^15^N, the fourth or fifth plants under each treatment were divided into ortets, spacers, ramets, and roots, dried in an oven at 70 °C for 48 h, and ground for ^15^N isotope composition analysis. The ^15^N/^14^N ratios were determined by isotope ratio mass spectrometer (DELTA V Advantage ThermoFisher Scientific, Inc., Waltham, MA, USA) according to the method of Yousfi et al. [64]. The abundance of the nitrogen isotope was expressed as atom percent (AP, atom %). The following calculations were performed for each plant component (root, spacer, ortet, and ramet).

AP excess ^15^N was calculated for each sample by subtracting the background ^15^N AP (0.366%) for each component from 15N AP in enriched samples. 

The percent of plant N derived from fertilizer (Ndff) was calculated as:%Ndff = (atom% ^15^N excess in plant sample / atom% ^15^N excess in fertilizer) ∗ 100,
where atom% ^15^N excess is the measured ^15^N content of the plant sample minus the background ^15^N content before fertilization. Total N accumulation (g) was calculated by N concentration and dry mass of each component. The accumulation of N that was assimilated by the plants from fertilizer-N was calculated using the following equations:N from fertilizer N (mg / plant) = Plant total N accumulation × Ndff plant /100 ∗ 1000

The N distribution was calculated as the ratio of the ^15^N accumulation amount (mg) in a given organ to the total ^15^N accumulation in the whole plant (mg).

### 4.5. Statistical Analysis

All characteristics are illustrated by means and standard deviation (mean ± SD). One-way analysis of variance (ANOVA) was performed using SPSS version 19.0 (Chicago, IL, USA). One-way ANOVA was used to evaluate the differences of growth traits, biomass accumulation, biomass allocation, nitrogen content, nitrogen allocation, ^15^N abundance, accumulation and distribution, and %Ndff among treatments and organs. Post hoc comparisons were performed using Duncan’s test at a significant level of *p* < 0.05. Before the analysis, all data were checked for normality and homogeneity of variances and log transformed to correct deviation from these assumptions when needed. A Kruskal–Wallis test was performed if the data were not normally distributed after the log transformation. We performed two-way ANOVAs with organ and treatments as fixed factors in our study to evaluate changes in N content and N allocation.

## 5. Conclusions

Irrespective of the forms of N, N supply had a significant effect on plant growth parameters, which facilitates buffalograss establishment. Moreover, buffalograss seedlings exhibited preferential uptake of nitrate under all nitrogen regimes. Nitrate promoted higher vegetation propagation, while the plants treated with this N source maintained greater biomass of spacers, ramets, aboveground, and total plant, and greater biomass allocation to ramets and N allocation to spacer, compared to the other N forms. Meanwhile, those plants supplied with the treatment +NH_4_NO_3_ displayed a significantly greater N content in the ramets, ^15^N abundance, and ^15^N accumulation in all organs. Although NH_4_^+^ can be used as a sole N source, it can result in intoxication symptoms in buffalograss, and +NO_3_ can alleviate NH_4_^+^-induced morphological disorders. Thus, nitrate should be preferred as an N source in buffalograss establishment from seeds, especially those habitats with nitrogen deficiency. This study suggests that current fertilization practices may be improved by modification of nitrogen form in fertilization practices.

## Figures and Tables

**Figure 1 plants-11-02086-f001:**
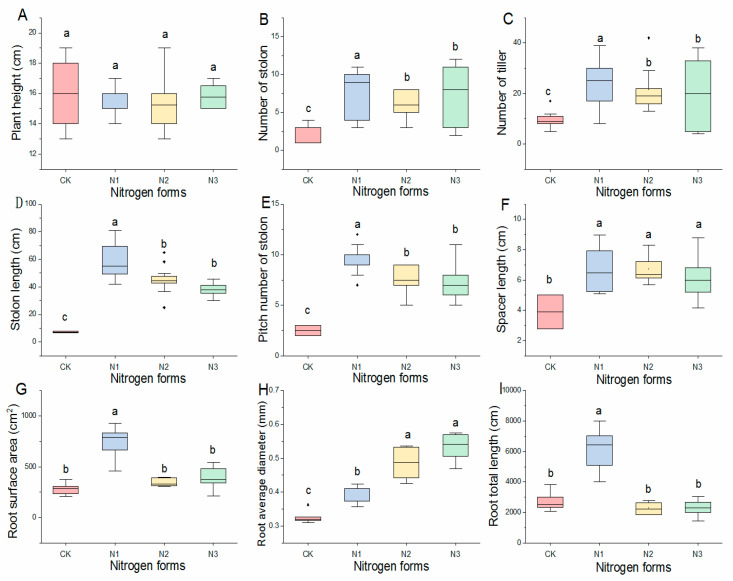
Box plot of growth characters of buffalograss seedlings, as affected by nitrate, ammonium, and ammonium nitrate. Each value is the mean ± SD (*n* = 10). The values not sharing the same letters are significantly different at *p <* 0.05 according to Duncan’s test of one-way ANOVA. CK, -NH_4_NO_3_; N1, +NO_3_^−^; N2, +NH_4_^+^; N3, +NH_4_NO_3_. Box plots indicate interquartile range in the box area, median (solid line in the box), mean (solid circle in the box) 25% and 75% percentiles (lower and upper box margins), 10% and 90% percentiles (lower and upper error bars), and outliers (solid rhombus outside the error bars). Note: (**A**): plant height of buffalograss seedlings in different nitrogen forms. (**B**): stolon number of buffalograss seedlings in different nitrogen forms. (**C**): tiller number of buffalograss seedlings in different nitrogen forms. (**D**): stolon length of buffalograss seedlings in different nitrogen forms. (**E**): pitch number of stolon of buffalograss seedlings in different nitrogen forms. (**F**): spacer length of buffalograss seedlings in different nitrogen forms. (**G**): root surface of buffalograss seedlings in different nitrogen forms. (**H**): root average diameterof buffalograss seedlings in different nitrogen forms. (**I**): root total length of buffalograss seedlings in different nitrogen forms.

**Figure 2 plants-11-02086-f002:**
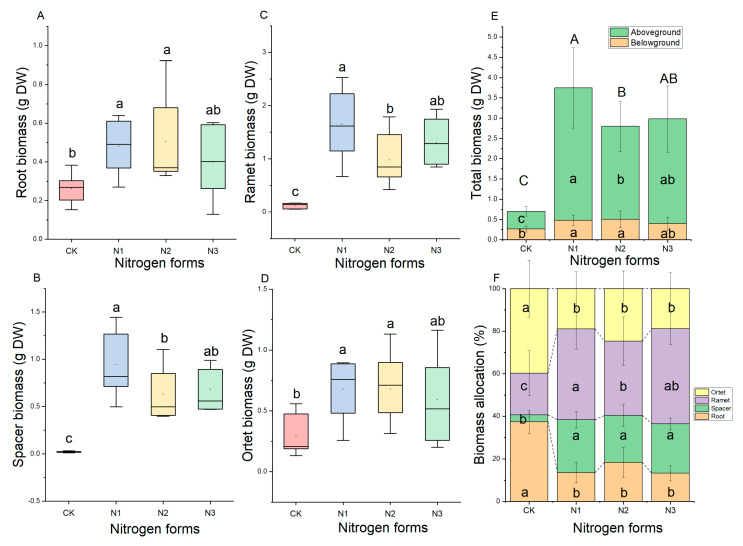
The influence of nitrogen form on biomass and biomass allocation of buffalograss seedlings. Each value is the mean ± SD (*n* = 6). CK, -NH_4_NO_3_; N1, +NO_3_^−^; N2, +NH_4_^+^; N3, +NH_4_NO_3_. Box plots indicate interquartile range in the box area, median (solid line in the box), mean (solid circle in the box) 25% and 75% percentiles (lower and upper box margins), 10% and 90% percentiles (lower and upper error bars), and outliers (solid rhombus outside the error bars). Note: (**A**): root biomass of buffalograss seedlings in different nitrogen forms. (**B**): spacer biomass of buffalograss seedlings in different nitrogen forms. (**C**): ramet biomass of buffalograss seedlings in different nitrogen forms. (**D**): orter biomass of buffalograss seedlings in different nitrogen forms. (**E**): total biomass of stolon of buffalograss seedlings in different nitrogen forms. (**F**): biomass allocation of buffalograss seedlings in different nitrogen forms. Different lowercase letters indicate a significant difference among different nitrogen forms (*p* < 0.05), values designated by different capital letters indicate significant difference of total biomass among nitrogen forms.

**Figure 3 plants-11-02086-f003:**
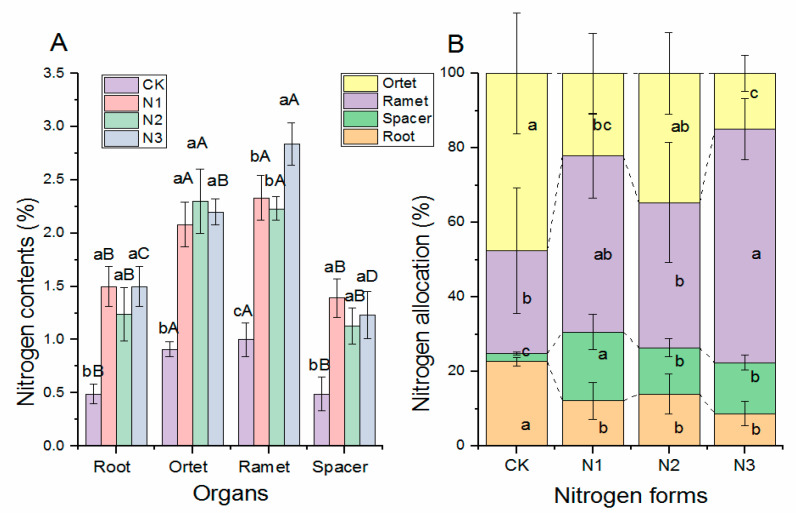
The influence of nitrogen form on N contents and N allocation of buffalograss seedlings. Each value is the mean ± SD (*n* = 5). CK, -NH_4_NO_3_; N1, +NO_3_^−^; N2, +NH_4_^+^; N3, +NH_4_NO_3_. The lowercase letters indicate significant difference at the 0.05 level among different treatments of the same organ; the capital letters indicate significant difference at the 0.05 level among different organs under a given treatment. Note: (**A**): nitrogen contents of different buffalograss seedlings organs in different nitrogen forms. (**B**): nitrogen allocation of different organs of buffalograss seedlings in different nitrogen forms. Different lowercase letters indicate a significant difference among different nitrogen forms (*p* < 0.05), Values designated by different capital letters indicate significant difference among organs.

**Figure 4 plants-11-02086-f004:**
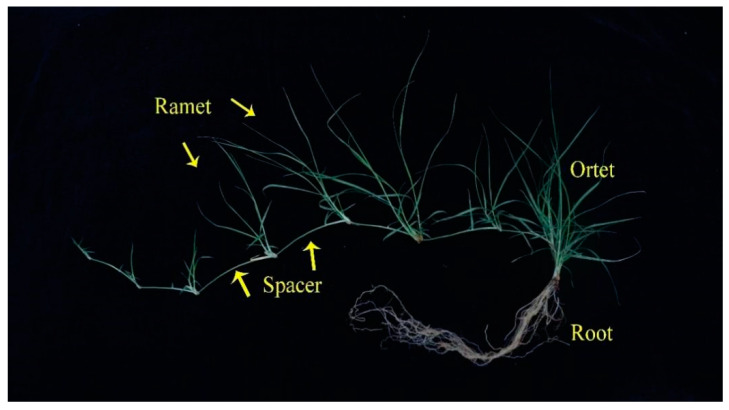
Schematic diagram of *Buchloe dactyloide.*

**Table 1 plants-11-02086-t001:** ANOVA interaction of treatments and tissues on N contents and N allocation in buffalograss seedlings.

Factors	*df*	N Contents		N Allocation	
		*F*	*p*	*F*	*p*
Organs (O)	3	144.162	<0.01	53.658	<0.01
Treatments (T)	3	120.894	<0.01	0.002	1
O ∗ T	9	5.627	<0.01	8.299	<0.01

**Table 2 plants-11-02086-t002:** ^15^N values and ^15^N allocation of buffalograss seedlings, as affected by nitrate, ammonium, and ammonium nitrate. Each value is the mean ± SD. The values not sharing the same letters are significantly different at *p* < 0.05 according to Duncan’s test of one-way ANOVAs. CK, -NH_4_NO_3_; N1, +NO_3_^−^; N2, +NH_4_^+^; N3, +NH_4_NO_3._

		Organs			
		Root	Spacer	Ramet	Ortet
^15^N abundance (atom %)	N1	4.22 ± 0.13 b	4.53 ± 0.23 b	4.40 ± 0.39 bB	5.09 ± 0.10 a
	N2	4.39 ± 0.48 b	4.94 ± 0.85 ab	4.89 ± 0.80 abAB	5.64 ± 0.60 a
	N3	4.66 ± 0.51 b	5.45 ± 0.86 ab	5.48 ± 0.63 abA	5.93 ± 0.72 a
Ndff (%)	N1	3.91 ± 0.13 b	4.22 ± 0.23 b	4.09 ± 0.40 bB	4.79 ± 0.11 a
	N2	4.08 ± 0.48 b	4.64 ± 0.86 ab	4.59 ± 0.82 abAB	5.35 ± 0.60 a
	N3	4.35 ± 0.52 b	5.15 ± 0.87 ab	5.19 ± 0.64 abA	5.65 ± 0.73 a
^15^N accumulation amount (mg/plant)	N1	3.45 ± 1.14	3.68 ± 1.00	3.54 ± 0.90	4.24 ± 1.46
	N2	2.52 ± 0.47 b	2.83 ± 0.37 ab	2.80 ± 0.32 ab	3.30 ± 0.52 a
	N3	3.41 ± 1.26	4.05 ± 1.58	4.07 ± 1.47	4.44 ± 1.67
^15^N distribution (%)	N1	22.98 ± 0.77 c	24.80 ± 0.66 b	23.99 ± 1.50 bc	28.23 ± 1.30 a
	N2	21.94 ± 1.00 c	24.78 ± 1.49 b	24.53 ± 0.91 b	28.75 ± 1.00 a
	N3	21.44 ± 0.81 c	25.24 ± 1.00 b	25.54 ± 0.88 b	27.78 ± 0.38 a

Note: The lowercase letters in the same row indicate significant difference at 0.05 level among different organs under given treatment, the capital letters in the same column indicate significant difference at 0.05 level among different treatments of same organ.

**Table 3 plants-11-02086-t003:** Nutrient solution composition of nitrogen form treatments.

Treatment (mM)	CK	N1	N2	N3
	-NH_4_NO_3_	+NO_3_^−^	+NH_4_^+^	+NH_4_NO_3_
KCL	1.25	0	1.5	0.25
CaCL_2_·6H_2_O	1.25	0	1.25	0
MgSO_4_·7H_2_O	0.5	0.5	0.5	0.5
KH_2_(PO_4_)	0.25	0.25	0	0
KNO_3_	0	1.25	0	1.25
Ca(NO_3_)_2_·4H_2_O	0	1.25	0	1.25
NH_4_H_2_PO_4_	0	0	0.25	0.25
NH_4_CL	0	0	3.5	3.5
H_3_BO_3_	0.0116	0.0116	0.0116	0.0116
MnCL_2_·4H_2_O	0.0046	0.0046	0.0046	0.0046
ZnSO_4_·7H_2_O	0.00019	0.00019	0.00019	0.00019
Na_2_MoO_4_	0.00012	0.00012	0.00012	0.00012
CuSO_4_·5H_2_O	0.00008	0.00008	0.00008	0.00008
FeSO_4_·7H_2_O	0.0125	0.0125	0.0125	0.0125
Na_2_EDTA	0.0125	0.0125	0.0125	0.0125

## Data Availability

Not applicable.

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
