# Peer review of "Effects of Nitrogen Forms on the Growth and Nitrogen Accumulation in Buchloe dactyloides Seedlings"

_plants, 2022, doi:10.3390/plants11162086_

Round 1

Reviewer 1 Report

Comments

The manuscript by Guo and collaborators aimed at evaluating the effects of different inorganic nitrogen forms (NO3--N, NH4+-N and NO3—N: NH4+-N = 1: 1) on the growth and nitrogen uptake in seedlings of buffalograss (Buchloe dactyloides). From their results, the authors concluded that applications of ammonium caused detrimental effects on buffalograss seedlings growth which could be alleviated by nitrate. The topic is interesting but as detailed in the following comments, I am not convinced by the results which are presented in the manuscript about the detrimental effects of N supplied as NH4+ compared to NO3-.

Comments

Whereas it is clear that the application of N increased plant growth compared to no N supply, I have several questions about the results presented in the manuscript.

- First, the means and the standard deviations given in the box plots (Fig 1, 2 and 3A) are very close and very high, respectively, and this is not in favor of significant differences ….

- Second, the effect of NH4 will depend on the concentration applied, thus it would have been necessary to compare the effects of several concentrations of N …

- In the experiments reported here, I was surprised by the very low amounts of 15N accumulated by the plants grown with either NH4 or NO3 … If I understood well, 30 mg of 15N were brought during the experiment.  However, from Table 2, I calculated that plants accumulated 3,75 mg15N/plant when grown with NO3 (N1) and 2,93 mg15N/plant when grown with NH4 (N2) and those values are very low … So my question is : where did go the applied 15N? Was it lixiviated? Did the authors control N concentrations in percolating solutions? On the other hand, I did not understand the sentence “The accumulation of 15N in the ramet was 17.5 ± 0.75, 1.16 ± 0.37, and 2.11±0.70 mg/plant in the N1, N2, and N3 treatments, respectively” (Lines 166-167). But these numbers do not correspond to the data given in Table 2.

- Also, still regarding the 15N experiment, the amount applied within N3 treatment is not clearly given. Was it 60 mg of 15N, ie 15 mg of each N source? Please precise.

Thus to conclude, I recommend that the authors read carefully their manuscript, checked carefully their data, including those given in the Tables ….  

Author Response

- First, the means and the standard deviations given in the box plots (Fig 1, 2 and 3A) are very close and very high, respectively, and this is not in favor of significant differences ….

A: Thank you for pointing this out. The data has no strict normal distribution, but the results don`t seem to make a difference when I try the Permutation multivariate analysis of variance (R ‘Vegan’). There still are significant difference among treatments.

- Second, the effect of NH4 will depend on the concentration applied, thus it would have been necessary to compare the effects of several concentrations of N …

A: Thank you for pointing this out. Actually, the plant was looked rather sickly in NH4 group, and it has the same manifestation when the clonal ramets of male and female buffalograss received the same treatments (N1-N4) in another sand culture experiments. We will test the effect of NH4 concentrations in buffalograss in our later experiments.

- In the experiments reported here, I was surprised by the very low amounts of 15N accumulated by the plants grown with either NH4 or NO3 … If I understood well, 30 mg of 15N were brought during the experiment.  However, from Table 2, I calculated that plants accumulated 3,75 mg15N/plant when grown with NO3 (N1) and 2,93 mg15N/plant when grown with NH4 (N2) and those values are very low … So my question is : where did go the applied 15N? Was it lixiviated? Did the authors control N concentrations in percolating solutions? On the other hand, I did not understand the sentence “The accumulation of 15N in the ramet was 17.5 ± 0.75, 1.16 ± 0.37, and 2.11±0.70 mg/plant in the N1, N2, and N3 treatments, respectively” (Lines 166-167). But these numbers do not correspond to the data given in Table 2.

A: We are very sorry for our negligence of missing and wrong information in Table 2, and we have modified them. We also found that the plant total N accumulation was wrong in our previous calculation, and have recalculated the accumulated N from fertilizer. The new results were displayed in Table 2, and description of Result, Conclusion and Abstract were modified according the update.  

The N from fertilizer N (mg / plant) = Plant total N accumulation × Ndff plant /100 * 1000

We applied marker containing 15N (like K15NO3, Ca(15NO3)2·4H2O etc.) instead of ordinary chemicals (like KNO3, Ca(NO3)2·4H2O, etc ) in nutrient solution when we did the 15N tracer treatment. Therefore, there were no problems in N concentration or others.

- Also, still regarding the 15N experiment, the amount applied within N3 treatment is not clearly given. Was it 60 mg of 15N, ie 15 mg of each N source? Please precise.

A: According to the comment, we have made correction in text.

***We have also revised what the reviewer did not mention in the previous manuscript. 

  • Somereferences have been modified and added for its necessary in text. In addition, we adjusted the number and format of references according to the requirements of the journal.
  • Some sentences in Abstract, Resultsand Conclusion has been rewritten according to other reviewer`s suggestions (part of 15N result).
  • We checkedand revised the text thoroughly for language 

Reviewer 2 Report

The present manuscript assesses the effects of nitrogen forms on the growth and nitrogen accumulation in the turfgrass Buchloe dactyloides. The study concluded with significant effects on several plant traits depending on the form of N used. The manuscript is clearly written with results correctly explained and covered in the discussion.

The main concern about the paper is the lack of information about why this studied grass is important. The main two points of the article are the study of the N forms (which a widely covered topic) and this turfgrass. The second main point is not correctly covered because in the introduction is being pointed that “it has become the most widely cultivated warm season turfgrass in northern China”; line 69; Why?. Why is also important to know the nitrogen necessities and effects of nitrogen of this plant?. From my point of view the importance of this plant should be more remarked to justify all the fertilization analysis and research done.

Proposed minor changes:

Figure 2: graph C the Y axis legend is “Niteogen”

Line 146: Is Table 1 instead of Table 2.

Table 3: why you have used different concentrations of NH4+ and NO3- (3.75 and 2.5 mM respectively)?

Author Response

The main concern about the paper is the lack of information about why this studied grass is important.

A: Thank you for pointing this out. We have modified and added the descriptions of the importance of buffalograss in the Introduction according to the comment.

Figure 2: graph C the Y axis legend is “Niteogen”

A: According to the comment, we have made correction in Figure 2.

Line 146: Is Table 1 instead of Table 2.

A: According to the comment, we have made correction in text.

Table 3: why you have used different concentrations of NH4+ and NO3- (3.75 and 2.5 mM respectively)?

A: N1 group include 1.25 mM KNO3 and 1.25 mM Ca(NO3)2·4H2O, then there is 3.75 [1.25+(1.25*2)] mM in total; while N2 group 0.25 mM NH4H2PO4 and 3.5 mM NH4CL, hence total N concentration of is 3.75 mM.

***We have also revised what the reviewer did not mention in the previous manuscript. 

  • Some references have been modified and added for its necessary in text. In addition, we adjusted the number and format of references according to the requirements of the journal.
  • Some sentences in Abstract, Results and Conclusion has been rewritten according to other reviewer`s suggestions (part of 15N result).
  • We checked and revised the text thoroughly for language 

Round 2

Reviewer 1 Report

The authors have  improved the manuscript. Therefore, I think that the revised manuscript could be accepted for publication.

Reviewer 2 Report

The authors have made all the proposed improvements so the manuscript can be accepted in the present form.